# Physicochemical Properties and Cellular Uptake of Astaxanthin-Loaded Emulsions

**DOI:** 10.3390/molecules24040727

**Published:** 2019-02-18

**Authors:** Xue Shen, Tianqi Fang, Jian Zheng, Mingruo Guo

**Affiliations:** 1College of Food Science and Engineering, Jilin University, Changchun 130062, China; shenxue417@163.com (X.S.); fangtianqi0527@126.com (T.F.); zhengjian85@jlu.edu.cn (J.Z.); 2Department of Food Science, Northeast Agriculture University, Harbin 150030, China; 3Department of Nutrition and Food Sciences, College of Agriculture and Life Sciences, University of Vermont, Burlington, VT 05405, USA

**Keywords:** astaxanthin, whey protein isolate, emulsifier, stability, bioavailability

## Abstract

Astaxanthin, a natural pigment carotenoid, is well known for its potential benefits to human health. However, its applications in the food industry are limited, due to its poor water-solubility and chemical instability. Six different emulsifiers were used to prepare astaxanthin-loaded emulsions, including whey protein isolate (WPI), polymerized whey protein (PWP), WPI-lecithin, PWP-lecithin, lecithin, and Tween20. The droplet size, zeta potential, storage stability, cytotoxicity, and astaxanthin uptake by Caco-2 cells were all investigated. The results showed that the droplet size of the emulsions ranged from 194 to 287 nm, depending on the type of emulsifier used. The entrapment efficiency of astaxanthin was as high as 90%. The astaxanthin-loaded emulsions showed good physicochemical stability during storage at 4 °C. The emulsifier type had a significant impact on the degradation rate of astaxanthin (*p* < 0.05). Cellular uptake of astaxanthin encapsulated into the emulsions was significantly higher than free astaxanthin (*p* < 0.05). Emulsion stabilized with WPI had the highest cellular uptake of astaxanthin (10.0 ± 0.2%), followed, in order, by that with PWP (8.49 ± 0.1%), WPI-lecithin (5.97 ± 0.1%), PWP-lecithin (5.05 ± 0.1%), lecithin (3.37 ± 0.2%), and Tween 20 (2.1 ± 0.1%). Results indicate that the whey protein-based emulsion has a high potential for improving the cellular uptake of astaxanthin.

## 1. Introduction

Astaxanthin is a xanthophyll carotenoid present in various microorganisms and marine animals [1]. Among the commercially important microalgae, *Haematococcus pluvialis* is considered to be the richest source of natural astaxanthin [2]. Astaxanthin cannot be synthesized de novo in the body by mammals and must be acquired from their diet [3]. Astaxanthin supplementation may have many benefits to human health. Astaxanthin has strong antioxidant activity [4], anticancer activity, anti-inflammatory activity [5,6], and can also reduce the risk of cardiovascular diseases [7], *Helicobacter pylori* infections [3], UV damage, immune dysfunctions, ageing, and age-related eye diseases [8]. Therefore, the utilization of carotenoids as nutraceutical ingredients into our foods is recommended [9].

The applications of astaxanthin in different food formulations are currently limited because of their poor water-solubility, high melting point, low bioavailability, and chemical degradation under certain conditions, like environmental acidity, heat, light, and oxygen [10]. Different strategies have been developed for improving its stability and bioavailability in foodstuffs, including microencapsulation [11,12], liposomes [13], emulsions [14], and nanodispersions [15]. In particular, astaxanthin in emulsions can be dissolved in the oil phase which, when created with a large surface area, facilitates the digestion of astaxanthin [16]. Oil-in-water emulsions can improve the storage stability and bioavailability of carotenoids [17]. Emulsifiers play an important role in emulsion stability and the bioavailability of carotenoids [18,19,20].

Whey proteins are amphiphilic molecules with high nutritional quality, surface activity, and emulsifying properties. Several studies have shown that whey proteins can be used as emulsifiers to stabilize carotenoid-loaded emulsions, such as β-carotene [21], lycopene [22], and lutein [23,24]. Whey proteins can form a protective layer around the droplets and protect the droplets from aggregation or coalescence as a result of electrostatic and hydrophobic interactions. Another advantage of whey proteins as an emulsifier or stabilizer is their antioxidant activity [25]. Interestingly, we recently found that the bioavailability of astaxanthin could be increased by the formation of astaxanthin nanodispersions with whey protein isolate (WPI) and/or polymerized whey protein (PWP) as stabilizers, suggesting that both WPI and PWP are good emulsifiers [26]. lecithin, a zwitterionic surfactant, has previously been shown to increase the cellular uptake of lutein in combination with whey protein hydrolysates [27]. Therefore, in this study, six different emulsifiers were used to prepare astaxanthin-loaded emulsions including WPI, PWP, WPI-lecithin, PWP-lecithin, lecithin and Tween20. The stability and cellular uptake of astaxanthin-loaded emulsions with six emulsifiers were systematically investigated.

## 2. Results and Discussion

### 2.1. Characterization of Astaxanthin-Loaded Emulsions

The mean droplet size and size distribution of astaxanthin-loaded emulsions stabilised by six emulsifiers (WPI, PWP, lecithin, WPI-lecithin, PWP-lecithin, and Tween 20) are shown in Figure 1A. All of the samples had monomodal distributions and submicron droplet sizes ranging from 193.87 ± 3.84 to 286.52 ± 19.75 nm, with a polydispersity index (PDI) value of less than 0.3. The mean droplet sizes of astaxanthin-loaded emulsions depended on the type of surfactant. The emulsion stabilized by lecithin showed a larger average droplet size than that of emulsions stabilized by WPI, which was in agreement with a previous study [28].

All of the astaxanthin-loaded emulsions were negatively charged, the magnitude of their electrical charge depending on the nature of the emulsifier used (Figure 1B). The absolute values of zeta potential were all higher than 50 mV in stabilized WPI, PWP, WPI-lecithin, PWP-lecithin, and lecithin emulsions, suggesting a stable colloid dispersion. Tween 20-stabilized emulsions had a significantly higher zeta potential value than WPI, PWP, WPI-lecithin, PWP-lecithin, and lecithin-stabilized emulsions (*p* < 0.05). Tween 20, a nonionic surfactant, may coat droplets with no charge. However, astaxanthin-loaded emulsions stabilized by Tween 20 had a negative zeta potential (−7.1 mV) at neutral pH, which may be due to the presence of free fatty acid impurities in the surfactant or medium-chain triacylglycerol (MCT) oil [29], or due to the preferential absorption of OH^-^ from H_2_O on the surface of the droplets [30].

The encapsulation efficiencies of astaxanthin were, in percentage, 96.9 ± 1.2, 97.2 ± 1.2, 90.7 ± 0.8, 90.1 ± 4.4, 89.7 ± 3.4, and 94.3 ± 1.1, respectively, for WPI-, PWP-, WPI-lecithin-, PWP-lecithin-, lecithin-, and Tween 20-stabilized emulsions (Figure 1C). The encapsulation efficiencies were all lower than 100%, indicating that some astaxanthin might be lost during the preparation of emulsions. Similar phenomena have also been observed by Liu et al. [31], who reported the loss of about 20% astaxanthin during emulsion preparation. The formation of free radicals as a result of acoustic cavitation during the homogenization process may partially contribute to the degradation or loss of carotenoids [32].

### 2.2. Storage Stability of Astaxanthin-Loaded Emulsions

#### 2.2.1. Effects of Storage Temperature on Physical and Chemical Stability of Astaxanthin-Loaded Emulsions

The change in particle size during storage at different temperatures (4, 25, 37, and 55 °C for 14 days) was relatively small (<10%) for astaxanthin-loaded emulsions stabilized by WPI (Figure 2A). The possible reason for the increase in particle size during storage in globular protein-stabilized emulsions was flocculation, as previously reported [33]. As the temperature increases, some non-polar groups located in the hydrophobic interior of the protein get exposed to the surface, which promotes the droplets flocculation through hydrophobic interactions [34]. The particle size during storage slightly increased (<5%) at 4 °C for all samples, indicating that low temperature storage was conducive to improving the physical stability of the emulsions, which was in accordance with the previous findings [35].

The astaxanthin content in WPI-stabilized emulsions decreased by 8%, 14%, 13%, and 20% at 5, 25, 37, and 55 °C for 14 days, respectively (Figure 2B). Changes in astaxanthin content with temperature increases showed a similar trend for all treated samples (data not shown). The astaxanthin retention was high in samples stored at 4 °C, indicating that a low temperature is critical to keep astaxanthin stable in the emulsions [34].

#### 2.2.2. Effects of Emulsifier Type on Physical and Chemical Stability of Astaxanthin-Loaded Emulsions

The changes in particle size were relatively small (<10%) during storage at 25 °C for 14 days for astaxanthin-loaded emulsions stabilized by WPI, PWP, WPI-lecithin, PWP-lecithin, and lecithin (Figure 3A). The droplet size of astaxanthin-loaded emulsions, stabilized by Tween 20, significantly increased from 193 to 212 nm during storage at 25 °C for 14 days (*p* < 0.05), which was likely attributed to some coalescence of droplets coated by a non-ionic surfactant [36]. Therefore, the choice of emulsifier provides a good opportunity to improve the physical stability of astaxanthin emulsions. In the current study, WPI provided the best physical stability of astaxanthin-loaded emulsions.

The emulsifier type had a pronounced impact on the astaxanthin retention in emulsions during storage at 25 °C for 14 days (pH 7.0) (Figure 3B). WPI- and PWP-stabilized emulsions showed a better protective effect from decreasing astaxanthin content than other emulsifiers. Mao et al. [37] reported that WPI-stabilized nanoemulsion shows the lowest degradation of β-carotene, which may be due to the antioxidant activity of whey proteins [25]. The majority of whey proteins are β-lactoglobulin (β-lg) and α-lactalbumin (α-la), which are generally responsible for the inhibition of lipid oxidation at the oil–water interface by scavenging free radicals through their cysteyl residues, disulfide bonds, and thiol functional groups [37,38].

### 2.3. Cytotoxicity of Astaxanthin-Loaded Emulsions

An MTT assay was used to evaluate the possible cytotoxicity of astaxanthin-loaded emulsions to Caco-2 cells at different dilutions in DMEM before Caco-2 cell uptake experiments (Figure 4). The cell viability of Caco-2 cells after incubation with 5-, 10-, and 100-fold dilutions of the six astaxanthin-loaded emulsions was >85%, >90%, and >94%, respectively, suggesting that six emulsions were nontoxic at the tested concentrations. A similar result was reported by [39], who found that the cell viability of Caco-2 cells was 85% after incubation with 3 mg/mL whey protein. Wei et al. [40] also reported that whey protein-, sodium caseinate-, and Tween 80-stabilized emulsions had no cytotoxicity toward Caco-2 cells. However, Frede et al. [41] reported that Tween 20-stabilized emulsions reduce the cell viability of HT-29 cells when incubated with the emulsions for 72 h. The contradiction between results might be caused by the variation in cell lines and incubation times.

### 2.4. Cellular Uptake of Encapsulated Astaxanthin in Emulsion Delivery Systems

Cellular uptake of astaxanthin in all samples showed a time-dependent behavior (Figure 5). Cellular uptake of astaxanthin encapsulated into the emulsions was significantly higher than free astaxanthin (*p* < 0.05), indicating an enhanced enterocyte cellular uptake of astaxanthin via the emulsion delivery system. The emulsifier type had a significant impact on the cellular uptake of astaxanthin by Caco-2 cells (*p* < 0.05). After 4 h of incubation, intracellular astaxanthin accumulation from astaxanthin-loaded emulsions stabilized by WPI, PWP, WPI-lecithin, PWP-lecithin, lecithin, and Tween 20 was 0.50, 0.42, 0.30, 0.25, 0.17, and 0.11 µg, which were respectively 10-, 8.4-, 6-, 5-, 3.4-, and 2.2-foldof that for free astaxanthin (0.05 µg). The droplet size and emulsifier type are generally considered to be the two possible reasons for the increase in cellular uptake of carotenoids in emulsions [14,40]. The smaller the droplet was, the higher the carotenoid uptake would be on the basis of the same emulsifier [40]. It has been reported that a decrease in particle size of microparticles increased the astaxanthin release from the microparticles in simulated gastric intestinal juice [42]. In this study, Tween 20-stabilized emulsions showed a significantly lower astaxanthin cellular uptake than that of emulsions stabilized by whey proteins, lecithin, and their combinations. Similarly, Tan et al. [43] reported that the cellular uptake of the lutein nanodispersion stabilized by sodium caseinate was higher than that which was stabilized by Tween 80. Thus, the type and nature of the emulsifiers may be a possible factor accounting for the cellular uptake efficiency. Richelle et al. (2001) reported that lycopene encapsulated in whey proteins (in a food-based formulations) exhibited similar lycopene bioavailability in plasma and buccal mucosa cells in humans, as that from tomato paste, suggesting that whey proteins might be used as carriers to deliver lycopene in a bioavailable and concentrated form [44]. Cellular uptake of astaxanthin in WPI-stabilized emulsions showed the highest cellular uptake of astaxanthin. The highest cellular uptake of β-carotene in WPI-stabilized emulsion was also observed by Lu et al. [40]. For oral consumption of emulsions, the droplets are passed through a model gastrointestinal system (mouth, stomach, small intestine, and intestinal mucosa) before being absorbed by enterocyte cells. The digestion within the gastrointestinal system may have a strong effect on protein-based emulsions. Therefore, in future work, it would be more appropriate to expose the emulsion droplets to simulated gastrointestinal tract prior to uptake assay in the Caco-2 cell.

## 3. Material and Methods

### 3.1. Materials

Whey protein isolate (WPI), containing 93.14% protein, 0.36% fat, 4.79% moisture, 1.6% ash, and 0.7% lactose, was purchased from Fonterra (Auckland, New Zealand). Lecithin (food grade) was purchased from HuaCheng Biological Technology Co. (Changchun, China). Tween 20, phosphate buffer solution (PBS), and dimethyl sulfoxide (DMSO) were purchased from Solarbio life science Co. (Beijing, China). Medium-chain triacylglycerol oil (MCT oil) was purchased from HuaCheng Biological Technology Co., (Changchun, China). Astaxanthin and its standard were purchased through Sigma (St. Louis, MO, USA). The content of *all-trans*-astaxanthin was 97.19%. Methanol, methyl *tert*-butyl ether (MTBE), acetone, and dichloromethane of HPLC gradient grade were purchased from Fisher Scientific Co. (Fair Lawn, NJ, USA), High-glucose and l-glutamine Dulbecco’s Modified Eagle Medium (DMEM) and fetal bovine serum (FBS) were purchased from Thermo Fisher Scientific (Gibco, Waltham, MA, USA). All other chemicals used were of reagent grade and purchased through Sigma (St. Louis, MO, USA). The water used in this study was filtered using a Millipore Milli-Q™ water purification system (Millipore Corp., Milford, MA, USA).

### 3.2. Preparation of Astaxanthin-Loaded Emulsions

Oil phase/astaxanthin was dissolved in MCT oil (25 mg/100 mL) by stirring (C-MAG HS7, IKA, Staufen, Germany) for 3 min at 40 °C, then for 40 min at room temperature, to ensure no visual astaxanthin crystallization existed. Nitrogen was filled in order to prevent the oxidation of astaxanthin during stirring. Aqueous phase/WPI solutions (2.5% (*w*/*v*)) as well as PWP solution (2.5% (*w*/*v*); 85 °C for 30 min) was prepared as previously reported [45]. Briefly, WPI solutions were prepared with deionized Milli-Q water and kept at 4 °C overnight to achieve complete hydration. The solutions were equilibrated at room temperature and were then adjusted to pH 7.0, followed by heating while constantly stirring with a magnetic stirring bar in a water bath. Lecithin and Tween 20 were also dissolved in Milli-Q water (2.5% (*w*/*v*)) using a magnetic stirrer (C-MAG HS7, IKA, Staufen, Germany). WPI/lecithin (1:1 (*w*/*v*)) and PWP/lecithin (1:1 (*v*/*v*)) solutions were also obtained by mixing the pre-prepared 2.5% solutions (WPI, PWP, and lecithin). The total amount of emulsifier (WPI+lecithin or PWP+lecithin) was equal in all six emulsions with 2.25 g/100 mL. Sodium azide (0.022% (*w*/*v*)) was added to the aqueous phase to inhibit the growth of the microorganism. The oil phase and aqueous phase were mixed at a ratio of 1:9 (*v*/*v*) and the coarse emulsion was formed using an Ultra-Turrax T25 high-speed blender (IKA, Staufen, Germany) at 12,000 rpm/min for 2 min. The coarse emulsion was then homogenized on ice using an ultrasonic processor (VCX800, vibra cell, Sonics, Newtown, CT, USA) with a 13 mm high grade titanium alloy probe at 40% amplitude for 5 min (10 s/5 s work/rest cycles). The container was wrapped up in tinfoil to minimize astaxanthin loss during treatment.

### 3.3. Determination of Particle Size and Zeta Potential

The mean particle size and zeta-potential (ζ) of emulsion droplets were determined by dynamic light scattering (DLS) and electrophoretic mobility (U_E_) using laser Doppler velocimetry technique and phase analysis light scattering and a Zetasizer Nano ZS 90 (Malvern Instruments, Malvern, UK). The samples were diluted 100-fold using deionized Milli-Q water. Then, 1 mL of diluted samples was transferred into the measuring cell. The refractive index (RI) values for oil droplets and water were 1.47 and 1.33, respectively. Particle size values were reported as Z-average (D_z_), which is the intensity-weighted mean hydrodynamic size of the particles. ζ was calculated based on the Henry equation. All measurements were conducted at 25 °C.

### 3.4. Extraction and Determination of Astaxanthin in Emulsions

Astaxanthin was extracted from emulsions using methanol and dichloromethane (1:1 (*v*/*v*)) as described [15], with some modifications. The sample (0.5 mL) was mixed with methanol/dichloromethane (2 mL) and vortexed for 10 min. The mixture was then subjected to centrifugation at 1000 *g* for 10 min. The upper colored supernatant containing astaxanthin was transferred to a 10 mL brown volumetric flask. The extraction was repeated with 1 mL dichloromethane three times until the aqueous layer was clear.

### 3.5. High Performance Liquid Chromatography (HPLC)

Astaxanthin analysis was performed using an UHPLC system (Nexera UHPLC LC-30A, Shimadzu, Kyoto, Japan) with PDA UV–VIS absorption detector, and a C30 reversed-phase analytical column (250 × 4.6 mm i.d., 5 μm, YMC, Co., Ltd, Kyoto, Japan) as previously reported [26]. The flow rate was set at 1.0 mL/min at room temperature. The injection volume was 20 μL, and the detection wavelength was 474 nm. The mobile phase was used as follows: solvent A, methanol; solvents B, MTBE; solvent C, phosphoric acid/H_2_O (1:99 (*v*/*v*)). The gradient elution of each sample was described as follows: initial, 81% A/15% B/4% C; 5 min linear changed to 66% A/30% B/4% C maintained for 10 min; 8 min linear changed to 16% A/80% B/4% C maintained for 4 min; 3 min linear changed to 81% A/15% B/4% C maintained for 5 min. The calibration of peak area versus astaxanthin standard concentration was linear in the measured concentration ranging from 0.1 μg/mL to 50 μg/mL for *all-trans*-astaxanthin (*R*^2^ > 0.99, *n* = 3). 

### 3.6. Storage Stability of Astaxanthin Emulsions

The emulsion samples were transferred into 50 mL centrifuge tubes immediately after preparation. The samples were stored at either 4, 25, 37, or 55 °C, in the dark for two weeks. Storage stability of emulsions was monitored by the change in particle size. The particle size of emulsion droplets was determined at 0, 7, and 14 days. Astaxanthin content was also determined as described [31], with some modification, every two days after emulsion formation. The emulsions were diluted 50 times in DMSO. An emulsion without astaxanthin diluted in DMSO was used as a blank. Absorbance measurement was performed at 474 nm using a microplate reader (Synergy HT, BioTek, Winooski, VT, USA). The calibration curve was prepared in the astaxanthin concentration (dissolved in DMSO) ranging from 0.1 to 50 μg/mL (*R*^2^ = 0.9991).

The retention of astaxanthin was calculated as follows:
Astaxanthin retention (%) = (C_t_/C_0_) × 100%,
where C_t_ = astaxanthin contents in the emulsions at storage time t and C_0_ = total content of astaxanthin added to the emulsion.

Encapsulation efficiency of astaxanthin was equivalent to astaxanthin retention after fresh emulsion formation by HPLC.

### 3.7. Determination of Caco-2 Cell Viability

To determine the cell viability after incubation with astaxanthin-loaded emulsions, MTT(-3-(4,5-dimethyl-2-thiazolyl)-2,5-diphenyl-2H-tetrazolium bromide) assay was carried out using Caco-2 cell (passages 30–40). Caco-2 cells were incubated in high-glucose and L-glutamine DMEM medium supplemented with 1% (*v*/*v*) penicillin/streptomycin and 10% (*v*/*v*) FBS. The cells were seeded at a density of 2.5 × 10^4^ cells/well in 96-well plates and incubated at 37 °C and 5% CO_2_ for 72 h. Culture medium was removed and 5-, 20-, and 100-fold dilutions of astaxanthin-loaded emulsions with cell culture medium added to the wells. DMEM without emulsions was used as a control. After 24 h incubation, the cells were washed with 200 μL PBS/well for three times. Two hundred microliters of MTT-containing medium (5 mg/mL MTT in DMEM) was added to each well. After 4 h incubation, the medium was removed and 150 μL DMSO was added to dissolve any formed formazan crystals. Absorbance measurement was performed at 570 nm using a microplate reader (Synergy HT, BioTek, Winooski, VT, USA). Relative cell viability (%) was calculated as the absorbance of nanoparticle cells after subtracting that of control cells [46].

### 3.8. Caco-2 Cellular Uptake Assay

Caco-2 cells were seeded in 6-well plates at a density of 5 × 10^5^ cells/well and incubated at 37 °C and 5% CO_2_ for 7 days as reported by Garrett et al. [47], with modifications. Astaxanthin-loaded emulsions were diluted to a final astaxanthin concentration of 5 μg/mL, whereas free astaxanthin in DMSO diluted with DMEM (5 μg/mL) was used as a control. The diluted emulsions (1 mL) were added to the wells and incubated at 37 °C and 5% CO_2_. The culture medium was removed and the cells were washed three times with PBS (4 °C). The cells were collected in 1 mL Hank’s buffer solution containing ethanol (10% (*v*/*v*)) and BHT (45 μmol/L), and then subjected to ultrasound (1 s/3 s) with the samples immersed in an ice–water bath during the treatment. The cells were collected at several time intervals (0, 1, 2, and 4 h), lysed, extracted, dried by nitrogen, redissolved in dichloromethane/acetone (4:6 (*v*/*v*)), and analyzed by HPLC.

### 3.9. Statistical Analyses

All experiments were performed in triplicates. Statistical analyses were carried out using the statistical program SPSS Version 17.0 SPSS (SPSS Inc. Chicago, IL, USA). One-way ANOVA was used to compare mean values. The LSD multiple comparisons post hoc test was used to determine *p* values. Results were presented as mean ± standard deviation (SD) and considered significantly different when *p* < 0.05.

## 4. Conclusions

WPI, PWP, WPI-lecithin, PWP-lecithin, lecithin, and Tween 20 can be used as emulsifiers to protect the astaxanthin loaded in emulsions and stabilize the overall astaxanthin-loaded emulsions. The cellular uptake of astaxanthin from the emulsions was improved in cell experiments. Emulsion stabilized with WPI showed the highest cellular accumulation of astaxanthin. The results suggested that the emulsion stabilized by whey protein delivery system remarkedly increased the cellular uptake of astaxanthin and potentially improved its bioavailability.

## Figures and Tables

**Figure 1 molecules-24-00727-f001:**
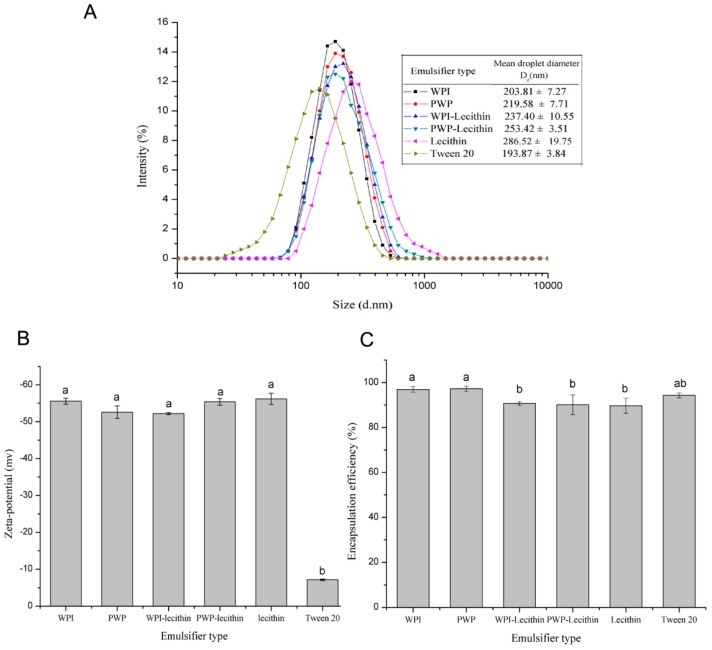
The effect of different emulsifiers on the mean droplet size and size distribution (**A**), zeta potential (**B**), and encapsulation efficiency (**C**) of astaxanthin-loaded emulsions at pH 7.0. All measurements were performed in triplicates. In the graphs, different letters (a,b) indicate significant differences (*p* < 0.05).

**Figure 2 molecules-24-00727-f002:**
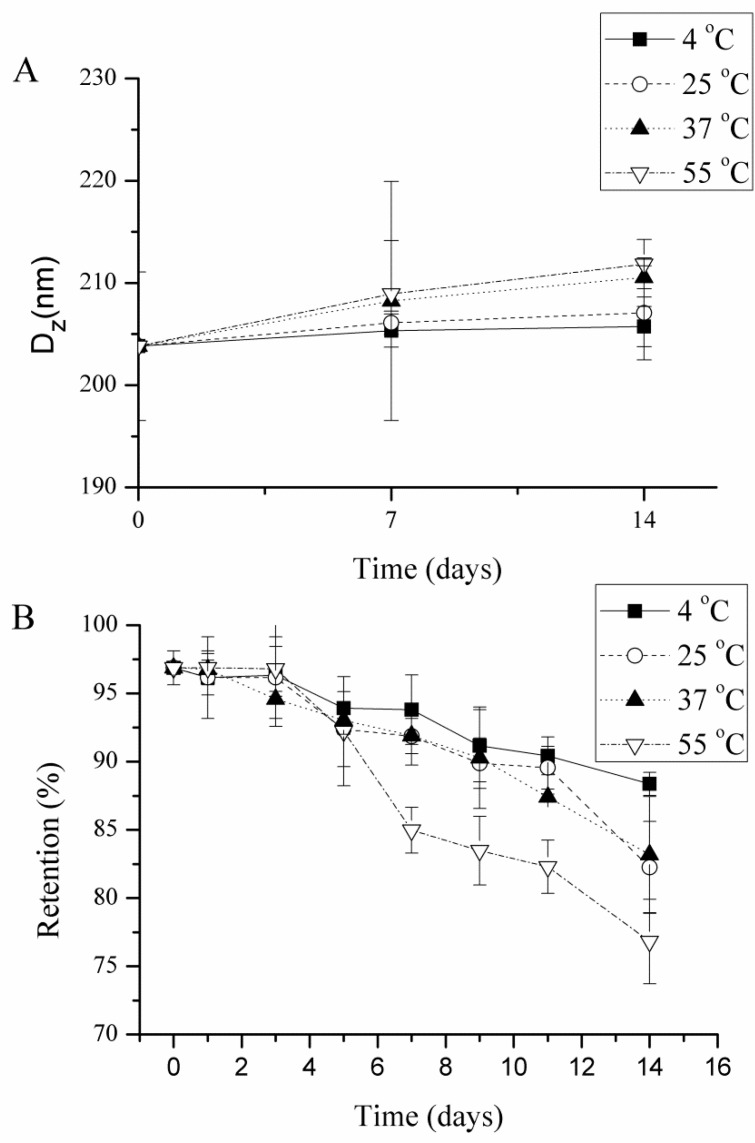
Influence of storage temperature on particle size (**A**) and the retention of astaxanthin (**B**) within emulsions stabilized by whey protein isolate (WPI) at pH 7.0. All measurements were performed in triplicates.

**Figure 3 molecules-24-00727-f003:**
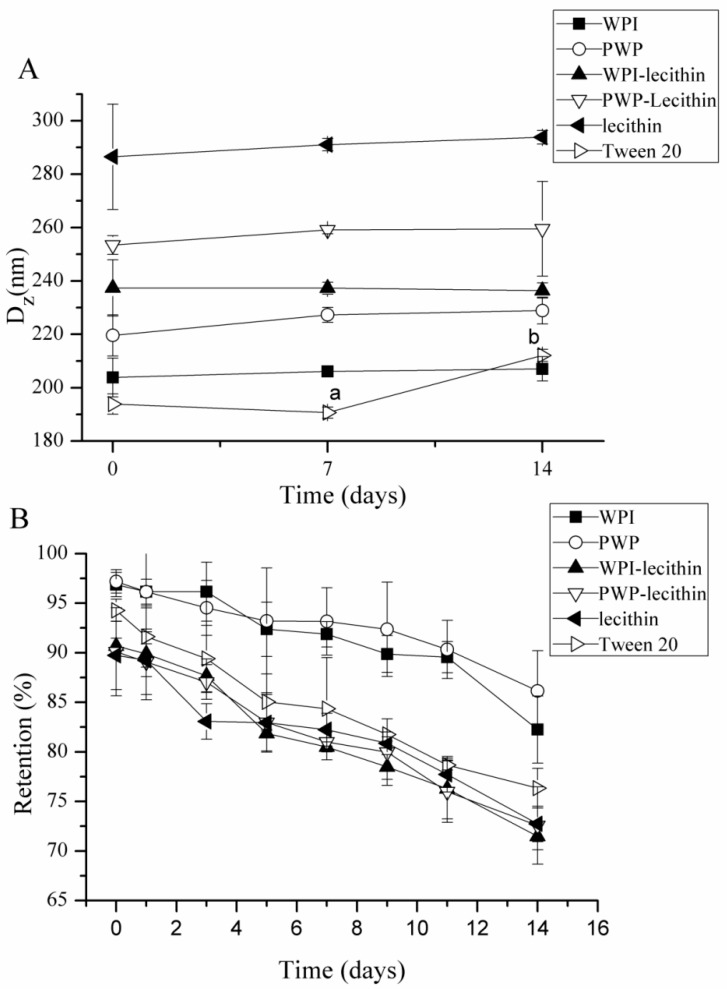
Influence of emulsifier type (WPI, polymerized whey protein (PWP), WPI-lecithin, PWP-lecithin, lecithin, and Tween 20) on particle size (**A**) and the retention of astaxanthin (**B**) within emulsions during storage (25 °C, pH 7.0). All measurements were performed in triplicates. In the graph, different letters (a,b) indicate significant differences (*p* < 0.05) between storage times, according to the Least Significant Difference (LSD) multiple range test.

**Figure 4 molecules-24-00727-f004:**
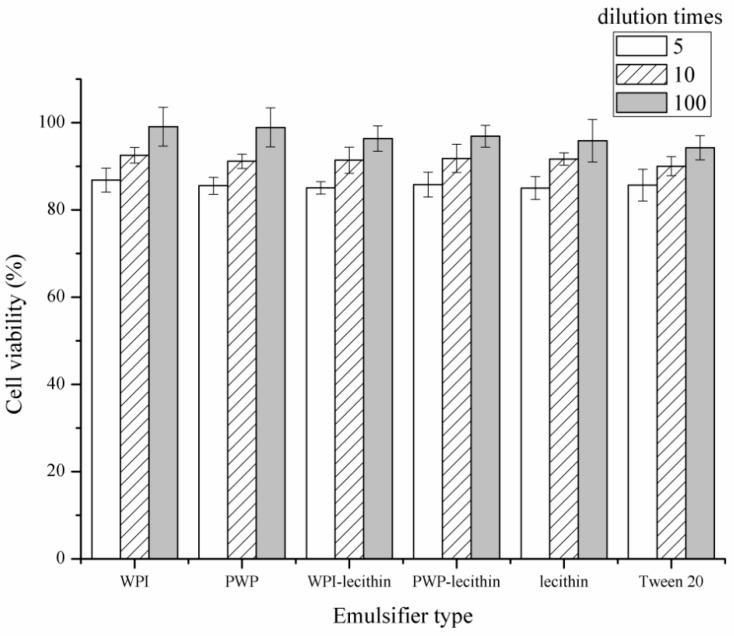
In vitro cytotoxicity of astaxanthin-loaded emulsions of different concentrations on Caco-2 cells by MTT assay. All measurements were performed in triplicates.

**Figure 5 molecules-24-00727-f005:**
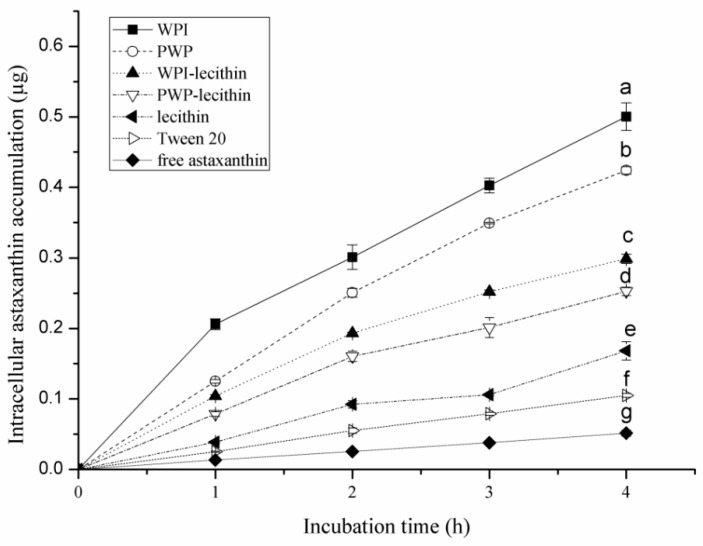
Astaxanthin uptake from astaxanthin-loaded emulsions with different emulsifiers by Caco-2 cells. All measurements were performed in triplicates. In the graph, different letters (a–g) indicate significant differences (*p* < 0.05) from other emulsifier types after 4 h incubation, according to the LSD multiple range.

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
