# Peer review of "Physicochemical Properties and Cellular Uptake of Astaxanthin-Loaded Emulsions"

_molecules, 2019, doi:10.3390/molecules24040727_

Round 1

Reviewer 1 Report

The submitted manuscript reports on Influence of different combinations of whey proteins and low molecular weight surfactants on the physicochemical properties and cellular uptake of astaxanthin-loaded emulsions stabilized. The approach is not difficult to understand and the findings will be useful in real applications in the field. The manuscript is almost well written, however, there are sections to be improved, particularly in the introduction and discussion, by focusing on the actual results and avoiding unrelated explanations.

General comments

I recommend adding low molecular weight surfactant to the title, the study includes both proteins and LMW surfactants.

I am wondering why the authors have not mentioned the finding of their previous work (Shen et al., 2018) with WPI and PWP. This could be something important to justify why the current study has been performed, based on the fact that the previous study did not include LMW surfactants. The introduction should show the important of the study, link it to previous studies and indicate what gap still needs to be satisfied.

Why not Tween 20-WPI or PWP combinations were performed?

Abstract

I would recommend starting the abstract with a sentence about why this study was realised.

Please indicate in the abstract the parameters studied

Line 17

Please change “and the its distribution was narrow” for “with a narrow size distribution”

“PDI” please spell it out

Line 19

“Good physicochemical stability” please be specific

“at low temperature” what temperature is it?

Introduction

The introduction should be more informative.

Line 30

“Among them” among microorganisms or marine animals? Be more concise….

Line 41

“including” is confusing, it might be both protective matrix or foodstuffs, please re-order and re-write sentence

Line 42-43

“which has a large surface and facilitates the digestion” this depends on the particle size that can be obtained, please change for “which, when created with a large surface area, facilitates the digestion of astaxanthin”

Line 44

Please combine this paragraph with the previous one

Line 47-48

“In addition to high nutritional quality, whey proteins are amphiphilic molecules with high 47 surface activity and emulsifying properties.” Please change for “Whey proteins are amphiphilic molecules with high nutritional quality, surface activity and emulsifying properties.”

Please mention in the introduction any previous studies on carotenoids stabilised emulsions using whey proteins

Line 48

“Proteins, especially whey proteins, can” please change for “whey proteins can…”

Line 50

“electrostatic and hydrophobic interactions”

Line 52

“were increased” change for “can be improved”

Line 52

“soluble aggregates also called” change for ”soluble aggregates, also called….”

Line 53

Give some more details about what the reference number [23] did and how emulsifying properties of whey proteins improved whey used in a polymerised form. Same for lecithin

Material and methods

Please add the complete information on the companies including city and country

Please give details on conditions used in preparation of protein dispersions (temperature, pH, did you allow time for full hydration)

laser Doppler velocimetry, make, model, city and country

Line 59-60

Please delete the repeated word “of”

Line 72

Which shear mixer used for stirring? Please mention type, model and manufacturer.

Line 75

How the whey protein solution was heated? Which system?

Based on what did you use this time/temperature combination?

Line 76

“Lecithin and Tween 20 were also dissolved in Milli-Q™ water” surfactants are soluble in oil, isn’t it? How they were dissolved in water?

Line 76-77

This sentence is not clear. Have you mixed the pre-prepared 2.5% solutions (WPI and Lecithin) to give a final concentration of 2.5% mix? Please make it clearer.

Line 81

An ultrasonic processor

Please provide city of manufacturer.

Line 88

Then, 1 mL……..

Line 92

Number of measurements and replicates

Line 94

Reference for extraction

Line 94-95

Protein solution preparation was already mentioned earlier, please delete…

Line 103

As previously reported

Line 104

Please delete “our previously reported”

Line 106

“MTBE” spell it out

Line 115

Please add “storage stability of emulsions was monitored by the change in particle size. The particle size……”

Line 123

2.7.Determination of Caco-2 cell viability

Line 126

“DMEM” “FCS” “PBS” “MTT” “DMSO” spell it out

Line 138

As reported by [28] with some modifications

Line 141-142

Combine into one single sentence

Results and discussion

Line 155

As measured by DLS

Line 157

“As a small molecule nonionic surfactant, Tween 20 can absorb to the oil-water interface…”, lecithin also is a surfactant and can do the same job. Why Tween emulsions got smaller size? Please give scientific reasons.

Line 167-168

Not sure what do you want to say by this sentence, I think it is repeated information and at same time, meaningless.

Line 170-172

Please make it clearer and more explained why Tween emulsions got a significantly different Z-potential value than other treatments. The explanation given is not satisfactory in my opinion.

Line 172

Encapsulation efficiency is not fully and well discussed

Line 177

“4, 25, 37 and 55 °C”

180-182

The authors need to talk about facts from previous studies in present tense, “As the temperature increases, some non-polar groups located in hydrophobic interior of protein get exposed to the surface, which promotes the droplets flocculation through hydrophobic interactions [35].”

Line 189

“was high in samples stored at 4oC”

Line 208-209

Would it be understood that replacing proteins by small molecular weight surfactant decreases the ability of the proteins to scavenge radicals and decreases the retention of carotenoids? Can you find more about this theory to discuss it better?

Line 226-227

“The contradiction of results might be caused by the variation in cell lines and incubation time”

Line 230-232

Please avoid repeating, same for figure 4 as well.

Line 249-250

Are you using the best resources to interpret your results? Would it be more suitable to cite only the work on globular protein in next sentence?

Line 260-261

“can be used as emulsifiers to protect the astaxanthin loaded in emulsions”

 Table 1

Particle size values and significance letters need revision, they do not seem to be statistically correct

Author Response

We appreciate it very much for the valuable comments and suggestions from you. The point-by-point response to the reviewer’s comments was attached.

Reviewer 2 Report

General aspects

The authors describe the investigation of astaxanthin loaded emulsions, regarding stability and cellular uptake. For this purpose, whey-protein isolate (WPI), polymerized whey protein, Lecithin and Tween20 are compared regarding their emulsion stability of incorporated astaxanthin. Following the formation of emulsions, stability test at room temperature are carried out for up to 14 days, and cellular uptake experiments conducted in a Caco-2 cell model of the intestinal barrier.

It was found that WPI resulted in a reasonably stable emulsion, had the smallest particle size (together with Tween20), and also resulted in hightest cellular uptake.

The manuscript is reasobly well written, and materials and methods are generally appropriate. The perhaps strongest disadvantage is that no digestion was carried out prior to testing of cellular uptake, however, digestion having a strong effect on protein-based emulsions, likely degrading most proteins and liberating astaxanthin. These limitations should be clearly mentioned in the discussion.  Further specific points are given below.  

Specific points

1.       Line 22 – absolute incorporations do not tell anything here – amount per what ? Better would be giving a percentage of incorporation, which could be easily given.

2.       Abstract – also p-values should be included, e.g. for cellular uptake experiments.

3.       Abstract – “cellular bioavailabiltiy” appears to be a rather unusual term. Better “cellular uptake”, in order to avoid confusion.

4.       Line 32 – better “may have many benefits” – as studies are generally either focussed on short-term surrogate endpoints or merely correlations.

5.       Line 35 – reference no. 8 is dealing with eye related diseases, so better to state “age-related eye diseases” or similar.

6.       Line 44-46 – also the review by Soukoulis (https://doi.org/10.1080/10408398.2014.971353) in Critical Rev Fd Sci Nutr highlights the relation of carotenoid emulsions and bioavailability and could be cited.

7.       Line 55 – please write out first time mentioned (PWP…).

8.       Line 56 – bioavailability was clearly not studied. Either refer to “cellular uptake” or “aspects of bioavailability”.

9.       Line 80 – give g-force instead of rpm.

10.   Line 103 – “Co” ??

11.   Line 111- please also add detection wavelength.

12.   2.8 – what was the amount of diluted emulsion placed into each well – 2 ml? In other words, what was the total amount of astaxanthin given on each well ? This should be mentioned, so percentage uptake can be calculated for these conditions, which is more insightful than just absolute amounts.

13.   Line 147 ff – please state if and how normality of distribution was assessed.

14.   Figure 1B – why does retention not start with 100% for time=0 ??

15.   Figure 1 heading: It is not known whether astaxanthin degrades or is simply “lost” from the emusions, so “chemical degradation” is the wrong term here. Just report retention in per cent. Same figure 2.

16.   Figure 2 – please check whether statistical interpretation (a,b denotation) is complete in figure.

17.   Line 250 and general – this is a strange way of starting a sentence. Author should be written out.

18.   Line 250-51 – but tomato past is not necessarily a product with a high bioavailability ? Thus, it is not accurate to report that this is the “best way” or similar to enhance bioavailability. Please change and tone down.

19.   Figure 4 – give percent uptake instead of absolute amounts, which is far less insightful.

20.   Figure 4 – please include statistical interpretation into figure as well.

21.   Discussion – another interesting article on astaxanthin is the one by Liu et al.: September 2016, Volume 11, Issue 3pp 302–310. This should also be cited.

22.   Line 262 – absorption also includes further transport across the basolateral side – thus better to stick to “cellular uptake”.

Author Response

Thank you for your message and feedback. We appreciate it very much for the valuable comments and suggestions from you. The point-by-point response to the reviewer’s  comments was attached.

Reviewer 3 Report

L64, Is Astaxanthin the all-trans-isomer? Several papers displayed that the emulsification efficiency of carotenoids and emulsion stability had big difference among the trans and cis isomer, e.g., Ono et al., J. Supercrit. Fluids, 138, 2018. Thus, the author should mention about that.

L118, Does the formula include astaxanthin isomers? Because when astaxanthin emulsion was storage at 55℃, the trans/cis isomerization would be promoted. 

L154-155, Please show typical particle size distribution of each emulsion if possible.

Author Response

(The authors gave the same response as above.)

Round 2

Reviewer 2 Report

Some further aspects could be included to foster readability and clarity. The statistical interpretation must be changed also. 

Line 24 - what was cellular uptake in % ?

Materials and methods: normally, the place of the company (town) should also be mentioned. 

Statistical analyses: It must be added how groups were compared - i.e. post-hoc test. Duncans does not truly correct for multiple comparison. Instead, Bonferroni or Tukeys must be used. Duncans is not acceptable.

Line 188 - units are missing !! Add at least percent. 

Figure titles - add number of replicates for each measurement.

Author Response

Response to Reviewer 2 Comments

Comments and Suggestions for Authors

Some further aspects could be included to foster readability and clarity. The statistical interpretation must be changed also. 

Response: Thanks for all your suggestions. We have made the amendment in the revised manuscript.

Line 24 - what was cellular uptake in % ?

Response: Thanks. Cellular uptake of astaxanthin values have been provided (ntracellular astaxanthin accumulation/ total addition*100%)

Original

Emulsion stabilized with WPI had a highest cellular uptake of astaxanthin, followed in order by that with PWP, WPI-Lecithin, PWP-Lecithin, Lecithin, and Tween 20.

Revised

Emulsion stabilized with WPI had a highest cellular uptake of astaxanthin (10.0 ± 0.2%), followed in order by that with PWP (8.49 ± 0.1% ), WPI-Lecithin (5.97 ± 0.1% ), PWP-Lecithin (5.05 ± 0.1%), Lecithin (3.37 ± 0.2% ), and Tween 20 (2.1 ± 0.1% ).

Materials and methods: normally, the place of the company (town) should also be mentioned. 

Response: Thanks for your suggestion. We have performed a thorough check-up carefully for the “Materials and methods” section and made the amendment as suggested.

Statistical analyses: It must be added how groups were compared - i.e. post-hoc test. Duncans does not truly correct for multiple comparison. Instead, Bonferroni or Tukeys must be used. Duncans is not acceptable.

Response: Thanks for your suggestion. We first analyzed the homogeneity of group variances using Levene's test. When Levene's test indicated homogenous variances among the groups,  LSD multiple comparisons post hoc test was used. In current version We have added the sentence “The LSD multiple comparisons post hoc test was used to determine P values.” to the Statistical analyses (section 2.9).

Line 188 - units are missing !! Add at least percent. 

Response: Thanks, we have added % in this sentence.

Original

The encapsulation efficiencies of astaxanthin were 96.9±1.2, 97.2±1.2, 90.7±0.8, 90.1±4.4, 89.7±3.4, and 94.3±1.1%, respectively, for WPI, PWP, WPI-Lecithin, PWP-Lecithin, Lecithin, and for Tween 20 stabilized emulsions (Fig. 1C).

Revised

The encapsulation efficiencies of astaxanthin were 96.9±1.2, 97.2±1.2, 90.7±0.8, 90.1±4.4, 89.7±3.4, and 94.3±1.1%, respectively, for WPI, PWP, WPI-Lecithin, PWP-Lecithin, Lecithin, and for Tween 20 stabilized emulsions (Fig. 1C).

Figure titles - add number of replicates for each measurement.

Response: Thanks for your suggestion. We have added one sentence “All measurements were performed in triplicates. ”for every figure.